# Learning for Dynamic Combinatorial Optimization without Training Data

## Abstract

We introduce DyCO-GNN, a novel unsupervised learning framework for Dynamic Combinatorial Optimization that requires no training data beyond the problem instance itself. DyCO-GNN leverages structural similarities across time-evolving graph snapshots to accelerate optimization while maintaining solution quality. We evaluate DyCO-GNN on dynamic maximum cut, maximum independent set, and the traveling salesman problem across diverse datasets of varying sizes, demonstrating its superior performance under tight and moderate time budgets. DyCO-GNN consistently outperforms the baseline methods, achieving high-quality solutions up to 3–60x faster, highlighting its practical effectiveness in rapidly evolving resource-constrained settings.

## 1 Introduction

Combinatorial optimization (CO) lies at the heart of many critical scientific and industrial problems (Papadimitriou & Steiglitz, 1982). Since most CO problems are NP-hard, solving large-scale instances is computationally prohibitive. Traditionally, solving such problems has relied on exact solvers, heuristics, and metaheuristics, whose design requires problem-specific insights. Recent advances in machine learning, particularly graph neural networks (GNNs), have opened new avenues for learning heuristics in a data-driven manner (Joshi et al., 2019; 2022; Gasse et al., 2019; Hudson et al., 2022; Karalias & Loukas, 2020; Bello et al., 2016; Khalil et al., 2017). GNNs have emerged as a powerful framework for learning over relational and structured data (Defferrard et al., 2016; Kipf & Welling, 2017; Hamilton et al., 2017; Veličković et al., 2018), with an inductive bias particularly suited for representing the underlying graph structures inherent in many CO problems.

Most existing approaches applying machine learning to CO are learning-heavy, requiring training on a large set of problem instances to learn *heuristics that generalize* to test instances. Training can take hours or even days on multiple GPUs (Wang & Li, 2023; Li et al., 2024). Schuetz et al. (2022) started another line of research that aims to develop an unsupervised learning (UL) method that learns *instance-specific heuristics*. Their method, PI-GNN, directly applies a GNN to the problem instance of interest and optimizes a CO objective. Under this design, no explicit "training" is involved; the runtime is the time it takes for the model to converge on each problem instance. Prior work has shown that problems such as maximum cut (MaxCut) and maximum independent set (MIS) on graphs with thousands of nodes can be solved by PI-GNN-based methods in minutes (Heydaribeni et al., 2024; Ichikawa, 2024).

While significant progress has been made in learning for static CO, many real-world problems are inherently dynamic, involving inputs or constraints that evolve over time (Yang et al., 2012; Zhang et al., 2021). In such settings, decisions must be updated continually, and practical algorithms must be efficient. A simple approach is to treat each snapshot of the problem instance as a static problem and solve it from scratch. However, significant overlap often exists in the node and edge sets across snapshots. Leveraging information from previous snapshots can improve both runtime and solution quality. For example, in MaxCut, the previous solution might serve as a good starting point after edge additions. Our proposed approach leverages such overlaps.

Specifically, we focus on learning for dynamic combinatorial optimization (DCO) and aim to advance the PI-GNN line of research. We present DyCO-GNN for DCO. To the best of our knowledge, our work is the first to apply machine learning to DCO problems. Our main contributions are as follows.

- We propose DyCO-GNN, a general UL-based optimization method for DCO that requires *no training data* except the problem instance of interest.
- We empirically validate the applicability of DyCO-GNN to various CO problems under dynamic settings, including MaxCut, MIS, and the traveling salesman problem (TSP).
- We demonstrate that DyCO-GNN achieves superior solution quality when the time budget is limited and outperforms the static counterpart with up to **3–60x** speedup.

## 2 RELATED WORK

Machine learning for static CO has predominantly relied on supervised learning approaches (Joshi et al., 2019; 2022; Vinyals et al., 2015; Gasse et al., 2019; Sun & Yang, 2023; Hudson et al., 2022; Li et al., 2023; 2024). These methods train models to predict high-quality solutions given a large dataset of problem instances paired with optimal or near-optimal labels. However, generating such labels is computationally expensive, especially for large-scale instances, making these methods less practical for many real-world applications.

To address this limitation, UL and reinforcement learning (RL) approaches have emerged as promising alternatives (Bello et al., 2016; Khalil et al., 2017; Kool et al., 2019; Karalias & Loukas, 2020; Qiu et al., 2022; Toenshoff et al., 2021; Tönshoff et al., 2023; Wang & Li, 2023; Sanokowski et al., 2023). These methods aim to learn optimization strategies directly from instance structures or reward signals without requiring labeled data. While they alleviate the need for ground-truth solutions, most UL and RL-based techniques still involve extensive offline training over large datasets to develop *heuristics that generalize*. Training such models can be time-consuming, often requiring hours or days on high-performance hardware (Wang & Li, 2023; Li et al., 2024).

A different paradigm was introduced by Schuetz et al. (2022), who proposed PI-GNN—an unsupervised framework that learns *instance-specific heuristics* by directly optimizing the CO objective on a single instance. This avoids any offline training and allows the method to be tailored to each problem instance during inference. PI-GNN combines a learnable embedding layer with a GNN and has been shown to achieve competitive performance across tasks such as MaxCut and MIS. Subsequent works have improved its solution quality and extended its design to incorporate higher-order relational reasoning (Heydaribeni et al., 2024; Ichikawa, 2024), achieving strong results even on large-scale graph instances.

Despite advances in instance-specific optimization, existing efforts have focused exclusively on static CO. The challenge of adapting learned heuristics efficiently to dynamic settings remains, to our knowledge, unexplored. Our work builds upon the PI-GNN line of research to propose new methods for DCO, where we aim to reuse and adapt learned solutions across temporally evolving problem instances without requiring optimization from scratch. We review the optimization theory literature on DCO, which is complementary to our learning-based approach, in Appendix A.

## 3 PRELIMINARIES

### 3.1 STATIC CO: PI-GNN AND QUADRATIC BINARY UNCONSTRAINED OPTIMIZATION (QUBO)

Consider a graph $G_p = (V_p, E_p, w_p)$ with node set $V_p = \{1, 2, \ldots, n_p\}$, edge set $E_p$, and edge weights $w_p$ representing or inherent in a CO problem instance $p$. PI-GNN is a general UL framework for static CO problems based on QUBO (Lucas, 2014; Glover et al., 2018; Djidjev et al., 2018). A QUBO problem is defined by:

$$\min_x \quad \ell(x; Q_p) = x^T Q_p x \tag{1}$$

where $x \in \{0, 1\}^N$ is a binary vector with $N$ components, and $Q_p \in \mathbb{R}^{N \times N}$ is a symmetric matrix encoding the cost coefficients, obtained based on the graph $G_p$. There are no explicit constraints, with all constraints incorporated implicitly via the structure of the Q matrix. Many CO problems, such as MaxCut, MIS, and TSP, can be reformulated as QUBO instances, making it a universal encoding framework for a wide class of problems. For a given static CO problem instance, PI-GNN learns to find a solution via $\mathcal{A}(G_p; \theta)$, where $\theta$ is the set of learnable parameters. Since the input graph has no node features, PI-GNN randomly initializes learnable embeddings for the nodes and feeds them to a

GNN. It then optimizes a differentiable QUBO objective in the form of Equation 1 by outputting a relaxed solution (i.e., $x \in [0, 1]^n$) followed by a final rounding step to obtain valid binary solutions.

**MaxCut.** Without loss of generality, we assume the edges have unit weights. The goal of MaxCut is to find a partition of the nodes into two disjoint subsets that maximizes the number of edges between them. We can model MaxCut solutions using $x \in \{0, 1\}^n$ where $x_i = 0$ if node $i$ is in one set, and $x_i = 1$ if node $i$ is in the other set. The objective can be formulated as $\sum_{(i,j) \in E}(2x_i x_j - x_i - x_j)$, which is an instance of QUBO (Glover et al., 2018).

**MIS.** MIS is the largest subset of non-adjacent nodes. Similar to MaxCut, we let $x_i = 1$ if node $i$ is in the independent set and $x_i = 0$ otherwise. The objective can be formulated as $-\sum_{i \in V} x_i + M \sum_{(i,j) \in E} x_i x_j$, where $M \in \mathbb{R}_{>0}$ is a penalty term enforcing the independent set constraint (Djidjev et al., 2018).

**TSP.** We consider symmetric TSP instances modeled as undirected complete graphs. TSP aims to find the route that visits each node exactly once and returns to the origin node with the lowest total distance traveled. In contrast to MaxCut and MIS, we use a binary matrix $X \in \{0, 1\}^{n \times n}$ to represent a route where $X_{ij} = 1$ denotes visiting node $i$ at step $j$. Following Lucas (2014) and The MathWorks Inc. (2024), we adopt the following objective: $\sum_{(i,j) \in E} w_{ij} \sum_{v=1}^{n} X_{iv} X_{j(v+1)} + M \sum_{i=1}^{n}(1 - \sum_{j=1}^{n} X_{ij})^2 + M \sum_{j=1}^{n}(1 - \sum_{i=1}^{n} X_{ij})^2$, where $w_{ij}$ is the distance between node $i$ and node $j$, $X_{j(n+1)} = X_{j1}$, accounting for the requirement of returning to the origin node, and $M \in \mathbb{R}_{>0}$ is a penalty term to ensure $X$ represents a valid route. During the QUBO objective computation, we simply flatten $X$ to an $n^2$-dimensional vector.

### 3.2 DCO on graphs

We focus on DCO problems on discrete-time dynamic graphs (DTDGs). A DTDG representing a DCO instance is a sequence $[G_p^1, G_p^2, \ldots, G_p^T]$ of graph snapshots where each $G_p^t = (V_p^t, E_p^t, w_p^t)$ has a node set $V_p^t = \{1, 2, \ldots, n_p^t\}$, an edge set $E_p^t$, and edge weights $w_p^t$. For dynamic MaxCut and MIS, we consider the scenario where the node set $V_p^t$ and edge set $E_p^t$ change over time, and $w_p^t = \mathbf{1}$ for all $t \in \{1, 2, \ldots, T\}$. For dynamic TSP, we consider the case where one of the nodes moves along a trajectory. Effectively, $V_p^1 = V_p^2 = \ldots = V_p^T$ and $E_p^1 = E_p^2 = \ldots = E_p^T$, but $w_p^t$ varies for each $t$.

Let $\Omega_p^t$ denote the set of discrete feasible solutions for snapshot $G_p^t$. For MaxCut and MIS, $\Omega_p^t$ is a subset of $\{0, 1\}^{n_p^t}$. For TSP, $\Omega_p^t$ is the set of all possible routes that visit each node exactly once and return to the origin node. The objective is to find the optimal solution for each snapshot of a given DCO problem instance: $x^{t*} = \operatorname{argmin}_{x \in \Omega_p^t} \ell(x; Q_p^t)$, where $\ell(\cdot; Q_p^t)$ denotes the cost function as defined earlier for each snapshot. For evaluation, we compute the mean approximation ratio (ApR) across all snapshots: Mean ApR $= \frac{1}{T} \sum_{t=1}^{T} \ell(x^t; Q_p^t)/\ell(x^{t*}; Q_p^t)$.

## 4 METHOD

### 4.1 WARM-STARTING PI-GNN

Given a sequence of graph snapshots $[G_p^1, G_p^2, \ldots, G_p^T]$, PI-GNN independently initiates an optimization process for each snapshot. One potential issue is that the time interval between two consecutive snapshots may be shorter than the convergence time of PI-GNN. In such cases, faster convergence is required. Considering the structural similarity across snapshots, a straightforward baseline method is to warm start the optimization using the parameters from the previous snapshot. Effectively, the "solution" to the previous snapshot serves as the initialization for the current one, allowing the model to "fine-tune" the solution instead of optimizing everything from scratch. We detail the procedure of warm-starting PI-GNN under different time budgets in Algorithm 1.

**Limitations of naive warm start.** Upon empirical evaluation, naively warm-starting PI-GNN exhibits several limitations. Figure 1 compares the performance of warm-started and static PI-GNN on dynamic MaxCut and MIS instances. Most notably, we observe that while warm start can potentially outperform static PI-GNN under a stringent time constraint, its advantages diminish quickly as the time budget increases. Specifically, when the time constraint is relaxed, warm-started models tend to produce lower-quality solutions than their static counterparts. Although warm start does accelerate convergence, the results it converges to are generally suboptimal.

**Algorithm 1** Warm-starting PI-GNN

**Require:**
Problem instance: $[G_p^1, G_p^2, \ldots, G_p^T]$
Max epochs for convergence: $\text{epoch}_{\text{max}}$
Epochs for warm start optimization: $\text{epoch}_{\text{ws}}$

1: Randomly initialize $\theta^1$
2: Construct $[Q_p^1, Q_p^2, \ldots, Q_p^T]$
   for $[G_p^1, G_p^2, \ldots, G_p^T]$
3: **for** $i = 1$ to $\text{epoch}_{\text{max}}$ **do**
4:     Predict $x^1$ via $\mathcal{A}(G_p^1; \theta^1)$
5:     Update $\theta^1$ using $\nabla \ell(x^1; Q_p^1)$
6: **end for**
7: **for** $t = 2$ to $T$ **do**
8:     $\theta^t \leftarrow \theta^{t-1}$
9:     **for** $i = 1$ to $\text{epoch}_{\text{ws}}$ **do**
10:        Predict $x^t$ via $\mathcal{A}(G_p^t; \theta^t)$
11:        Update $\theta^t$ using $\nabla \ell(x^t; Q_p^t)$
12:    **end for**
13: **end for**

**Algorithm 2** DyCO-GNN

**Require:**
Problem instance: $[G_p^1, G_p^2, \ldots, G_p^T]$
Max epochs for convergence: $\text{epoch}_{\text{max}}$
Epochs for warm start optimization: $\text{epoch}_{\text{ws}}$
SP parameters: $\lambda_{\text{shrink}}$ and $\lambda_{\text{perturb}}$

1: Randomly initialize $\theta^1$
2: Construct $[Q_p^1, Q_p^2, \ldots, Q_p^T]$
   for $[G_p^1, G_p^2, \ldots, G_p^T]$
3: **for** $i = 1$ to $\text{epoch}_{\text{max}}$ **do**
4:     Predict $x^1$ via $\mathcal{A}(G_p^1; \theta^1)$
5:     Update $\theta^1$ using $\nabla \ell(x^1; Q_p^1)$
6: **end for**
7: **for** $t = 2$ to $T$ **do**
8:     $\theta^t \leftarrow \lambda_{\text{shrink}} \theta^{t-1} + \lambda_{\text{perturb}} \epsilon^t$
9:     **for** $i = 1$ to $\text{epoch}_{\text{ws}}$ **do**
10:        Predict $x^t$ via $\mathcal{A}(G_p^t; \theta^t)$
11:        Update $\theta^t$ using $\nabla \ell(x^t; Q_p^t)$
12:    **end for**
13: **end for**

Figure 1: Performance of static PI-GNN and warm-started PI-GNN on dynamic MaxCut and MIS instances. Detailed setup is explained in Section 5.

These phenomena are closely tied to the role of initialization in deep learning (Sutskever et al., 2013; Glorot & Bengio, 2010; He et al., 2015) and can be attributed to the highly nonconvex nature of the optimization objective with respect to the model parameters, which gives rise to numerous local optima. When the first snapshot is optimized to convergence, the model undergoes extensive optimization, resulting in increased confidence in its predictions. Consequently, when the model is warm-started, the gradients become small due to this overconfidence, making it more difficult for the model to escape local optima. This hampers its ability to discover higher-quality solutions despite faster initial progress.

### 4.2 DyCO-GNN: FAST CONVERGENCE AND ROBUST SOLUTION QUALITY

To address the shortcomings of naively warm-starting PI-GNN, we propose DyCO-GNN, a simple yet effective method that facilitates both fast convergence and robust solution quality across graph snapshots. Unlike standard warm start, which reuses the exact model parameters from the previous snapshot, DyCO-GNN integrates a strategic initialization method, shrink and perturb (SP), originally proposed in (Ash & Adams, 2020) in a different context for supervised learning tasks. The goal is to retain the benefits of accelerated convergence while mitigating the tendency of the model to become trapped in suboptimal local minima due to overconfident predictions caused by warm start.

SP was originally designed to close the generalization gap caused by naively warm-starting neural network training. It shrinks the model parameters and then adds perturbation noise. Shrinking the parameters effectively decreases the model's confidence while preserving the learned hypothesis. Adding noise empirically improves training time and test performance. It is important to note that there is no notion of generalization in our method as the model parameters are independently optimized for each individual problem instance snapshot. Despite the fact that SP tackles a different problem, we successfully adopt it to resolve the issues of warm-started PI-GNN.

When a new snapshot $G_p^t$ arrives, DyCO-GNN applies SP to the warm-started parameters before starting the optimization process. The newly initialized parameters are defined as

$$\theta^t \leftarrow \lambda_{\text{shrink}}\theta^{t-1} + \lambda_{\text{perturb}}\epsilon^t,$$

where $0 < \lambda_{\text{shrink}} < 1$, $0 < \lambda_{\text{perturb}} < 1$, and $\epsilon^t \sim \mathcal{N}(0, \sigma^2)$. With the reintroduced gradient diversity and disrupted premature overconfidence, DyCO-GNN effectively escapes the local minima in the loss landscape. This gentle destabilization promotes exploration of alternative descent paths, functioning as a principled soft reset mechanism that facilitates more effective and robust optimization.

With this simple design, DyCO-GNN achieves a balanced trade-off: it preserves the efficiency of warm start under tight time constraints while enabling the model to reach higher-quality solutions as more computational budget becomes available. The pseudocode of DyCO-GNN is provided in Algorithm 2. As we show in Section 5, DyCO-GNN consistently outperforms warm-started PI-GNN across DCO problems such as dynamic MaxCut, MIS, and TSP, particularly under moderate to generous time budgets where naive warm start fails to find high-quality solutions effectively. We will also show that DyCO-GNN finds better solutions than the converged static PI-GNN much faster.

### 4.3 THEORETICAL SUPPORT

We analytically support the advantage of SP over warm start. The theorem is based on the Goemans & Williamson (1995) (GW) algorithm, the best-known optimization algorithm for solving MaxCut. The GW algorithm is structured similar to DyCO-GNN, but with a more computationally intensive QUBO relaxation (an SDP) and rounding step (randomized projections). We adopt it to show that the advantage of SP persists even if we advance the relaxation and rounding steps.

**Theorem 1.** *Fix the solution $X_0$ of the GW SDP step, and define $X_\lambda := Proj_{\mathcal{X}}(X_0 + \lambda Z)$, where $\lambda \in \mathbb{R}_{>0}$, $Z$ is a symmetric random matrix sampled from the Gaussian Orthogonal Ensemble, and $Proj_{\mathcal{X}}(\cdot)$ is projection onto set $\mathcal{X} = \{X |\ X \succeq 0,\ diag(X) = 1\}$. Denote the GW rounding step by $R : \{\mathcal{X}, \Omega\} \to \{0, 1, \dots, c^*\}$, where $\Omega$ is the random seed set of the cut plane and $c^*$ is the maximum achievable cut size. Let $\mathcal{C}_{opt} = \{X \in \mathcal{X} :\ \mathbb{P}_\Omega(R(X, \omega) = c^*) > 0\}$. Assume that the set $C_{opt}$ has positive Lebesgue measure in $\mathcal{X}$, and that $X_0 \notin \mathcal{C}_{opt}$. Then, there exists a $\lambda > 0$ such that*

$$\mathbb{P}_{\Omega,Z}(R(X_\lambda, \omega) = c^*) > \mathbb{P}_\Omega(R(X_0, \omega) = c^*) = 0 \ .$$

In words, the theorem shows that (pre-rounding) perturbations can strictly increase the probability of finding the optimal cut. The proof is shown in Appendix E, along with a detailed description of the GW algorithm, and a corollary extending the result to perturbations of the GW SDP initialization.

## 5 EXPERIMENTS

We empirically evaluate DyCO-GNN on dynamic instances of MaxCut, MIS, and TSP of varying problem sizes. Baselines include static PI-GNN and warm-started PI-GNN. We exclude methods that rely on extensive offline training across instance distributions. This choice aligns with the core design philosophy of our proposed approach, which, like PI-GNN, emphasizes *instance-specific adaptability* over generalization. While generalizable methods trained on large datasets may offer strong performance on similar distributions, prior research has shown that methods from the PI-GNN family often achieve comparable performance across static CO problems like MaxCut and MIS (Schuetz et al., 2022; Wang & Li, 2023; Ichikawa, 2024). Moreover, the generalization gap of the methods with generalizability increases significantly when distribution shift is present (Karalias & Loukas, 2020; Wang & Li, 2023). By focusing our comparison on methods within the PI-GNN family, we provide a clearer assessment of the algorithmic innovations without conflating results with the (dis)advantages of dataset-level generalization. We also compare with the results of non-neural baselines in Appendix D.3.

### 5.1 DATASETS

We utilized established graph datasets that contain temporal information to ensure that the dynamic evolution of the graphs accurately reflects realistic, time-dependent structural changes. This approach allows us to capture authentic patterns of change within the graph over time, thereby enhancing the

Table 1: Mean ApR on dynamic MaxCut. Values closer to 1 are better ($\uparrow$). All methods use GCNConv. "emb", "GNN", and "full" refer to applying SP to the embedding layer, GNN layers, and all layers, respectively. The best result for each time budget is in bold. The first time each method surpasses the converged solution of static PI-GNN is highlighted.

| | | Time budget (epochs/seconds) | | | | | | | |
| --- | --- | --- | --- | --- | --- | --- | --- | --- | --- |
| | Method | 50/0.06 | 100/0.12 | 200/0.23 | 300/0.34 | 500/0.56 | 1000/1.11 | 2000/2.21 | 3000/3.32 |
| Infectious | PI-GNN (static) | 0.82906 | 0.92646 | 0.97203 | 0.98509 | 0.98801 | 0.98871 | **0.98887** | **0.98888** |
| | PI-GNN (warm) | **0.93460** | 0.95681 | 0.96874 | 0.97170 | 0.97186 | 0.96980 | 0.97196 | 0.96959 |
| | DyCO-GNN (emb) | 0.88562 | 0.96018 | 0.97817 | 0.98458 | 0.97695 | 0.97874 | 0.98117 | 0.96473 |
| | DyCO-GNN (GNN) | 0.93187 | 0.96485 | 0.97962 | 0.98366 | 0.98478 | 0.98456 | 0.98460 | 0.98409 |
| | DyCO-GNN (full) | 0.86286 | **0.96830** | **0.98384** | **0.98793** | **0.98947** | **0.98878** | 0.98869 | 0.98836 |
| | | Time budget (epochs/seconds) | | | | | | | |
| | Method | 50/0.06 | 100/0.12 | 200/0.24 | 300/0.36 | 500/0.59 | 1000/1.18 | 2000/2.36 | 3000/3.53 |
| UC Social | PI-GNN (static) | 0.97557 | 0.99485 | 0.99764 | 0.99798 | 0.99809 | 0.99817 | 0.99820 | 0.99823 |
| | PI-GNN (warm) | 0.99363 | 0.99455 | 0.99469 | 0.99492 | 0.99501 | 0.99519 | 0.99515 | 0.99519 |
| | DyCO-GNN (emb) | **0.99634** | 0.99709 | 0.99734 | 0.99720 | 0.99711 | 0.99710 | 0.99711 | 0.99705 |
| | DyCO-GNN (GNN) | 0.99352 | 0.99714 | 0.99779 | 0.99781 | 0.99786 | 0.99782 | 0.99782 | 0.99779 |
| | DyCO-GNN (full) | 0.99404 | **0.99724** | **0.99825** | **0.99841** | **0.99843** | **0.99848** | **0.99846** | **0.99848** |
| | | Time budget (epochs/seconds) | | | | | | | |
| | Method | 50/0.14 | 100/0.29 | 200/0.57 | 300/0.85 | 500/1.41 | 1000/2.82 | 2000/5.63 | 3000/8.44 |
| DBLP | PI-GNN (static) | 0.80083 | 0.97292 | 0.98905 | 0.98973 | 0.99017 | 0.99054 | 0.99081 | 0.99095 |
| | PI-GNN (warm) | 0.98804 | 0.98890 | 0.98957 | 0.98969 | 0.98976 | 0.99001 | 0.99009 | 0.99039 |
| | DyCO-GNN (emb) | 0.98858 | 0.98995 | 0.99028 | 0.99039 | 0.99044 | 0.99047 | 0.99061 | 0.99041 |
| | DyCO-GNN (GNN) | 0.98950 | 0.99110 | 0.99145 | 0.99155 | 0.99164 | 0.99176 | 0.99180 | 0.99183 |
| | DyCO-GNN (full) | **0.99319** | **0.99515** | **0.99524** | **0.99515** | **0.99509** | **0.99495** | **0.99484** | **0.99477** |

validity and applicability of our experimental results. We summarize the statistics of the original and preprocessed datasets in Table 4 in Appendix B.

For MaxCut and MIS, we employed Infectious (Isella et al., 2011), a human contact network; UC Social (Opsahl & Panzarasa, 2009), a communication network; and DBLP (Ley, 2002), a citation network. All datasets are publicly available in the Koblenz Network Collection (Kunegis, 2013). Each dataset is a graph with timestamps attached to all edges. For preprocessing, we began by sorting all edge events in chronological order, followed by the removal of self-loops and duplicate edges. Finally, we made all graphs undirected. To construct DTDG snapshots for MaxCut, we converted the graphs into 10 snapshots by linearly increasing the number of edges to include in each of them (i.e., the dynamic graph grows over time). For Infectious and UC Social, we added ~10% of total edges of the final graph every step; for DBLP, we added ~2% each step, considering its total number of edges is much larger. Edge additions will lead to MIS constraint violations. In that case, a postprocessing step that reuses previous solutions and removes violations would be sufficient to solve the DCO problem. Therefore, to construct DTDG snapshots for MIS, we reversed the snapshot order and turn edge additions into edge deletions.

For TSP, we first took static TSP benchmark problems from TSPLIB (Reinelt, 1991). Specifically, we considered burma14, ulysses22, and st70. We added an extra node and let it move along a straight line within the region defined by the other existing nodes. We recorded the coordinates of 5 equally spaced locations along the trajectory (i.e., 5 snapshots in total).

## 5.2 IMPLEMENTATION DETAILS

DyCO-GNN consists of a node embedding layer and two graph convolution layers. Unlike previous works (Schuetz et al., 2022; Heydaribeni et al., 2024; Ichikawa, 2024) that used different embedding and intermediate hidden dimensions for each problem instance, we used 512 for the embedding dimension and 256 for the hidden dimension in all our experiments. We experimented with the graph convolution operator (GCNConv) from Kipf & Welling (2017) and the GraphSAGE operator (SAGEConv) from Hamilton et al. (2017). We found that DyCO-GNN with GCNConv failed to find good enough solutions to TSP, so we only report results of the SAGEConv models on dynamic TSP. We set $\lambda_{\text{shrink}} = 0.4$ and $\lambda_{\text{perturb}} = 0.1$ without further tuning. All MaxCut and MIS experiments were repeated five times; all TSP experiments were repeated ten times. We obtained the ground truth for each snapshot by formulating it as a QUBO problem and solving it using the Gurobi solver (Gurobi Optimization, LLC, 2024) with a time limit of 60 seconds. Additional implementation details are provided in Appendix C.

Table 2: Mean ApR on dynamic MIS. Values closer to 1 are better ($\uparrow$). All methods use GCNConv. "emb", "GNN", and "full" refer to applying SP to the embedding layer, GNN layers, and all layers, respectively. The best result for each time budget is in bold. The first time each method surpasses the converged solution of static PI-GNN is highlighted.

|  | | Time budget (epochs/seconds) | | | | | | | |
|---|---|---|---|---|---|---|---|---|---|
|  | Method | 50/0.06 | 100/0.12 | 200/0.23 | 300/0.34 | 500/0.56 | 1000/1.12 | 2000/2.22 | 3000/3.33 |
| Infectious | PI-GNN (static) | 0.67757 | 0.70395 | 0.72587 | 0.73944 | 0.78275 | 0.86258 | 0.87577 | 0.87651 |
|  | PI-GNN (warm) | 0.50072 | 0.50217 | 0.50255 | 0.50285 | 0.50845 | 0.50620 | 0.50381 | 0.50309 |
|  | DyCO-GNN (emb) | **0.78941** | **0.79577** | 0.77912 | 0.76798 | 0.74713 | 0.72983 | 0.72221 | 0.70964 |
|  | DyCO-GNN (GNN) | 0.69457 | 0.74386 | **0.79747** | **0.78105** | 0.75516 | 0.73314 | 0.71517 | 0.71508 |
|  | DyCO-GNN (full) | 0.69461 | 0.72760 | 0.75278 | 0.77278 | **0.84044** | ==**0.88254**== | 0.88397 | **0.88515** |

|  | | Time budget (epochs/seconds) | | | | | | | |
|---|---|---|---|---|---|---|---|---|---|
|  | Method | 50/0.06 | 100/0.12 | 200/0.24 | 300/0.36 | 500/0.60 | 1000/1.19 | 2000/2.37 | 3000/3.56 |
| UC Social | PI-GNN (static) | 0.50103 | 0.60923 | 0.74196 | 0.82713 | 0.89385 | 0.91340 | 0.91594 | 0.91655 |
|  | PI-GNN (warm) | 0.73879 | 0.74222 | 0.74561 | 0.74828 | 0.74731 | 0.74740 | 0.75011 | 0.75016 |
|  | DyCO-GNN (emb) | **0.87396** | **0.86808** | **0.85321** | 0.84667 | 0.83863 | 0.82997 | 0.82259 | 0.81732 |
|  | DyCO-GNN (GNN) | 0.68646 | 0.78990 | 0.82860 | 0.82898 | 0.82632 | 0.82217 | 0.81881 | 0.81739 |
|  | DyCO-GNN (full) | 0.54819 | 0.66512 | 0.80108 | **0.87445** | **0.91600** | ==**0.92037**== | **0.92069** | **0.91984** |

|  | | Time budget (epochs/seconds) | | | | | | | |
|---|---|---|---|---|---|---|---|---|---|
|  | Method | 50/0.14 | 100/0.28 | 200/0.57 | 300/0.85 | 500/1.40 | 1000/2.80 | 2000/5.60 | 3000/8.41 |
| DBLP | PI-GNN (static) | 0.18364 | 0.48119 | 0.89690 | 0.93320 | 0.94304 | 0.94636 | 0.94766 | 0.94813 |
|  | PI-GNN (warm) | 0.93136 | 0.93313 | 0.93459 | 0.93516 | 0.93595 | 0.93672 | 0.93771 | 0.93925 |
|  | DyCO-GNN (emb) | ==**0.95517**== | **0.95599** | **0.95529** | 0.95491 | 0.95478 | 0.95398 | 0.95446 | 0.95448 |
|  | DyCO-GNN (GNN) | 0.73970 | 0.93442 | 0.94082 | 0.94114 | 0.94119 | 0.94135 | 0.94169 | 0.94183 |
|  | DyCO-GNN (full) | 0.26570 | 0.65637 | ==0.95175== | **0.96550** | **0.96895** | **0.96972** | **0.97016** | **0.97042** |

## 5.3 EMPIRICAL RESULTS ON DCO PROBLEMS

Tables 1, 2, and 3 report the mean ApR of all methods on dynamic MaxCut, MIS, and TSP, respectively. We also evaluate variants of DyCO-GNN that apply SP to different layers of DyCO-GNN. For MaxCut and MIS, results are reported using the final model checkpoint. For TSP, both the final and the best-performing checkpoints are considered. The wall-clock time of the best-performing checkpoint includes the total decoding time over all checkpoints available. As noted in Section 4, the advantage of warm-started PI-GNN compared to static PI-GNN diminishes quickly when we relax the runtime constraint. DyCO-GNN closes this performance gap and consistently outperforms static PI-GNN and warm-started PI-GNN across all problems and datasets. Even on a strict budget (first column of each table), DyCO-GNN surpasses warm-started PI-GNN in most cases. Crucially, DyCO-GNN consistently finds better solutions than fully converged static PI-GNN, and often within just **1.67%** to **33.33%** of the total runtime. This improvement possibly stems from information carryover and continual learning on a similar graph structure, enabled by the utilization of the previously learned hypothesis. Although the three versions of DyCO-GNN give strong results, no single setting dominates all tasks and datasets. We will discuss the implications in Section 6. We visualize snapshot-level ApRs under different time budgets in Figures 2, 3, and 4. Notably, DyCO-GNN maintains its edge across almost all evaluated snapshots, confirming its robustness. Additional experimental results are provided in Appendix D.

**Sensitivity of different methods to varying degrees of change.** Tables 1 and 2 show that the quality gap of converged solutions between static PI-GNN and DyCO-GNN (resp. warm-started PI-GNN) is larger (resp. smaller) for DBLP. This is because we constructed the DTDG for DBLP with a smaller relative change in the edge set ($\sim$2% of total edges), which leads to greater structural overlaps. Consequently, warm-started PI-GNN and DyCO-GNN would be stronger in such a scenario. We further conducted a sensitivity analysis by varying the degrees of change in the DTDGs. Results are illustrated in Figure 5. When $\Delta$edges is getting smaller, the gap between static and warm-started PI-GNN is narrowed, and the advantage of DyCO-GNN compared to static PI-GNN is more salient, which is consistent with our hypothesis.

**Sensitivity analysis on SP parameters.** To demonstrate the generality and robustness of our method without relying on dataset-specific tuning, we deliberately chose a single, reasonable set of SP parameter values (i.e., $\lambda_{\text{shrink}}$ and $\lambda_{\text{perturb}}$) and applied it uniformly, highlighting that strong performance can be achieved without sensitive hyperparameter tuning. For completeness, we include a sensitivity analysis on the SP parameters in Appendix D.4.

Table 3: Mean ApR on dynamic TSP. Values closer to 1 are better (↓). The best-performing checkpoint was taken. "emb", "GNN", and "full" refer to applying SP to the embedding layer, GNN layers, and all layers, respectively. The best result for each time budget is in bold. The first time each method surpasses the converged solution of static PI-GNN is highlighted.

| | | Time budget (epochs/seconds) | | | | | | |
|---|---|---|---|---|---|---|---|---|
| | Method | 500/0.31 | 1000/0.62 | 2000/1.24 | 3000/1.86 | 5000/3.10 | 10000/6.21 | - |
| burma14 | PI-GNN (static) | 1.31234 | 1.13584 | 1.06970 | 1.05998 | 1.04756 | 1.03358 | - |
| | PI-GNN (warm) | 1.28388 | 1.28313 | 1.25798 | 1.23433 | 1.21120 | 1.13860 | - |
| | DyCO-GNN (emb) | 1.15106 | 1.13302 | 1.09256 | 1.07814 | 1.06478 | 1.04782 | - |
| | DyCO-GNN (GNN) | **1.10365** | **1.06370** | 1.05406 | 1.05668 | 1.03737 | **1.01811** | - |
| | DyCO-GNN (full) | 1.15884 | 1.10839 | **1.04531** | **1.03134** | **1.03160** | 1.04032 | - |
| | | Time budget (epochs/seconds) | | | | | | |
| | Method | 500/0.32 | 1000/0.64 | 2000/1.27 | 3000/1.90 | 5000/3.18 | 10000/6.37 | - |
| ulysses22 | PI-GNN (static) | 1.76783 | 1.48249 | 1.26151 | 1.17969 | **1.11249** | 1.08867 | - |
| | PI-GNN (warm) | **1.18266** | 1.17228 | 1.18642 | 1.18381 | 1.17534 | 1.19563 | - |
| | DyCO-GNN (emb) | 1.20391 | **1.16486** | 1.16819 | 1.15758 | 1.14706 | 1.13088 | - |
| | DyCO-GNN (GNN) | 1.25758 | 1.19970 | **1.14971** | **1.12991** | 1.12469 | 1.13545 | - |
| | DyCO-GNN (full) | 1.30026 | 1.27399 | 1.22204 | 1.16843 | 1.12538 | **1.08516** | - |
| | | Time budget (epochs/seconds) | | | | | | |
| | Method | 500/0.43 | 1000/0.85 | 2000/1.70 | 3000/2.54 | 5000/4.23 | 10000/8.46 | 20000/16.95 |
| st70 | PI-GNN (static) | 2.01621 | 1.98448 | 1.92607 | 1.82142 | 1.59433 | 1.43824 | 1.36945 |
| | PI-GNN (warm) | 1.47258 | 1.46055 | 1.42443 | 1.42589 | 1.38696 | **1.36187** | 1.34213 |
| | DyCO-GNN (emb) | **1.45563** | **1.42486** | 1.38436 | **1.36485** | 1.35266 | 1.30753 | 1.29239 |
| | DyCO-GNN (GNN) | 1.53036 | 1.44924 | **1.36227** | **1.33620** | 1.32063 | 1.27857 | **1.23655** |
| | DyCO-GNN (full) | 1.56795 | 1.54532 | 1.38231 | **1.34845** | **1.30566** | **1.27127** | 1.24412 |

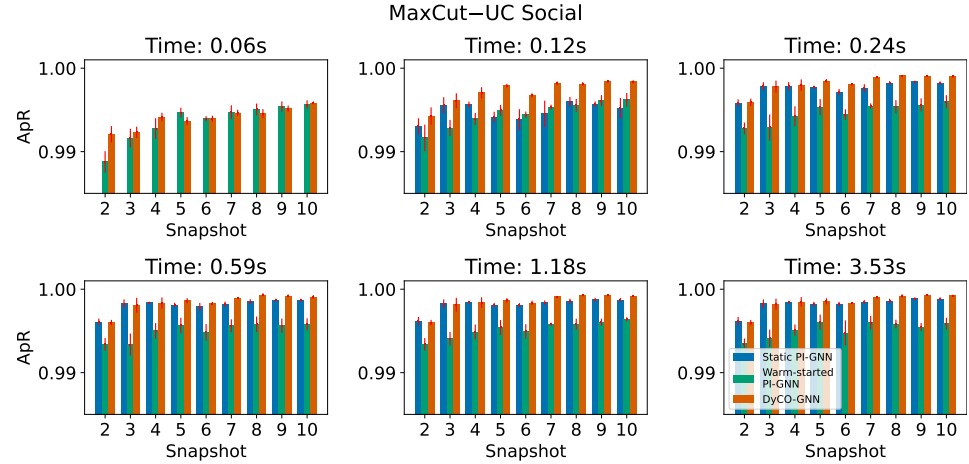

Figure 2: Snapshot-level ApRs on dynamic MaxCut instance (UC Social). All methods use GCNConv.

## 6  DISCUSSION

**Conclusion.** We introduced DyCO-GNN, the first learning-based framework designed to solve DCO problems. Through extensive experiments on dynamic MaxCut, MIS, and TSP, we demonstrated that DyCO-GNN consistently outperforms both static and warm-started PI-GNN baselines under varying runtime constraints and across diverse problem settings. Our analysis reveals that DyCO-GNN achieves superior solution quality even under strict time budgets, always surpassing the best-performing checkpoints of the baseline methods in a fraction of their runtime. These improvements are significant in rapidly changing environments, where timely decision-making is critical. Furthermore, DyCO-GNN's ability to efficiently adapt to evolving problem instance snapshots without optimization from scratch or external supervision underscores its potential for real-world deployment in dynamic, resource-constrained settings.

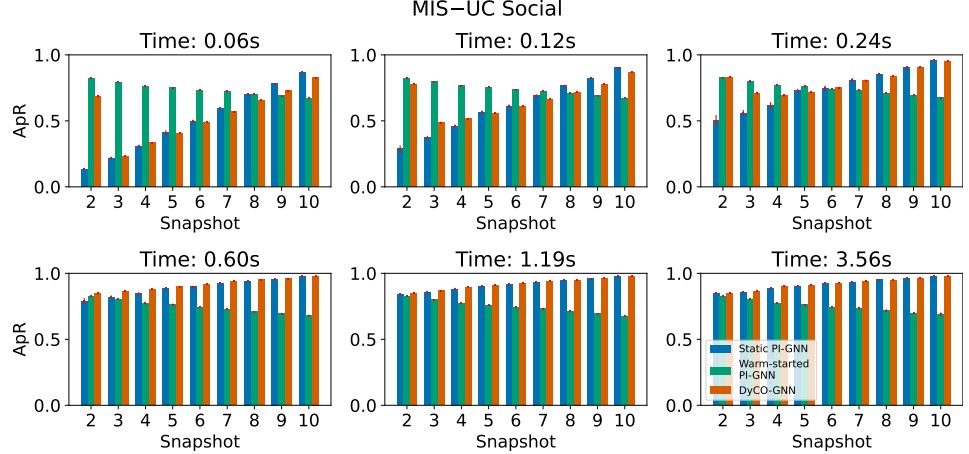

Figure 3: Snapshot-level ApRs on dynamic MIS instance (UC Social). All methods use GCNConv.

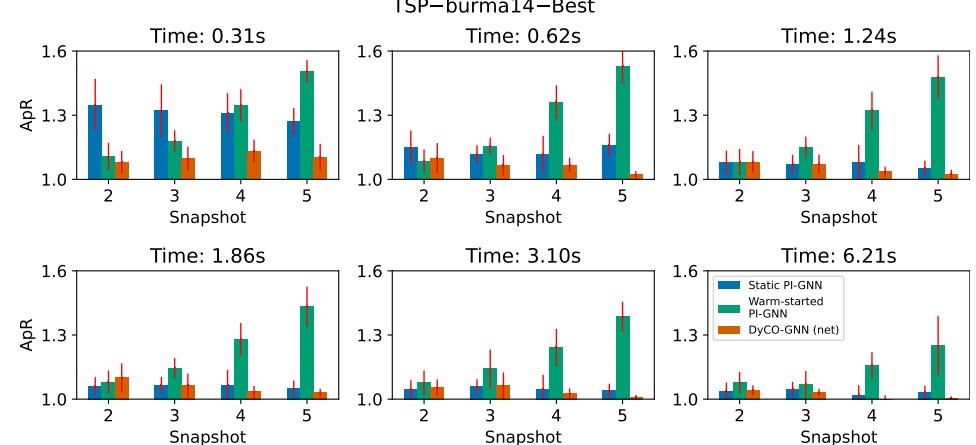

Figure 4: Snapshot-level ApRs on dynamic TSP instance (burma14). The best checkpoint was taken.

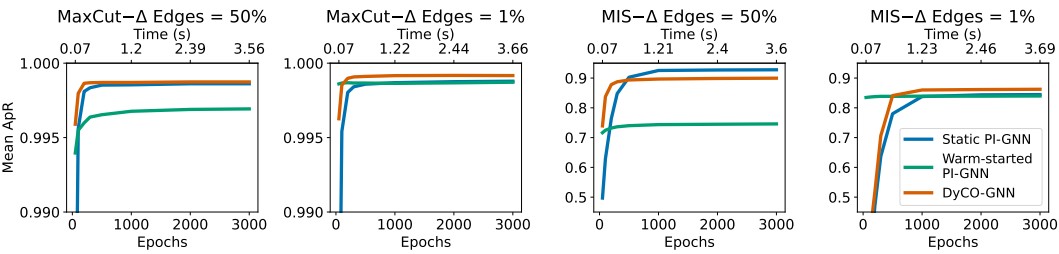

Figure 5: Sensitivity analysis using UC Social.

**Limitations and future work.** We have shown that applying SP to different layers of DyCO-GNN yields varying performance. A promising direction for future work is to make the SP step adaptive. Currently, the embedding layer and the GNN layers are jointly optimized without explicit decoupling. By introducing appropriate regularization strategies or auxiliary loss functions, we can promote distinct functional roles across layers; for instance, encouraging the embedding layer to capture representations specific to both the problem and graph structure and the GNN layers to learn to aggregate such representations. Additionally, if we evaluate the likelihood of changes in node assignments under dynamic conditions based on certain heuristics, it becomes possible to apply SP in a node-wise, adaptive manner.

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

## A    ADDITIONAL RELATED WORK

DCO problems have been addressed from a theoretical perspective (Onak & Rubinfeld, 2010; Thorup, 2007), including for the dynamic MaxCut (Wasim & King, 2020), MIS (Assadi et al., 2018), and TSP (Ausiello et al., 2009) problems, as well as from the perspective of designing heuristics (Yang et al., 2012). Our work brings a learning-based approach to this line of work.

The idea of warm-starting semidefinite programs in general, and the starting solutions of interior-point methods (which can be used to solved SDPs, among other problem classes) has been explored from a theoretical perspective (Yildirim & Wright, 2002; John & Yıldırım, 2008; Angell & Mccallum, 2024). Our theoretical findings complement this literature by exploring the impacts of warm-starting SDP solutions for the MaxCut problem. Our learning-based method DyCO-GNN also brings similar ideas into the realm of learning-based DCO methods.

## B    DATASET STATISTICS

Table 4: Statistics of all datasets. Original edges can be directed and have duplicates.

| Dataset | Original Nodes | Original Edges | Preprocessed Nodes | Preprocessed Edges | $\Delta$ edges |
|---|---|---|---|---|---|
| Infectious | 410 | 17,298 | 410 | 2765 | $\sim$276 (10%) |
| UC Social | 1899 | 59,835 | 1899 | 13,838 | $\sim$1384 (10%) |
| DBLP | 12,590 | 49,759 | 12,590 | 49,636 | $\sim$993 (2%) |
| burma14 | 14 | 91 | 15 | 105 | - |
| ulysses22 | 22 | 231 | 23 | 253 | - |
| st70 | 70 | 2,415 | 71 | 2,485 | - |

## C    ADDITIONAL IMPLEMENTATION DETAILS

We used the Adam optimizer (Kingma & Ba, 2015); the learning rate was set to 0.001 for all MaxCut and MIS experiments, 0.0002 for burma14 and st70, and 0.0005 for ulysses22. As described in Algorithms 1 and 2, we optimize over the first snapshot for $\text{epoch}_{max}$ epochs. Thus, we skip the first snapshot during evaluation since the results will be identical across all methods. We set $\text{epoch}_{max} = 3000$ for all MaxCut and MIS experiments, $\text{epoch}_{max} = 10000$ for burma14 and ulysses22, and $\text{epoch}_{max} = 20000$ for st70.

The penalty term $M$ was set to 2 for MIS and $2 \times \max_{(i,j) \in E} w_{ij}$ (i.e., 2×the max distance between any two nodes) for TSP. For MIS, we postprocessed the output of our model by greedily removing violations. For TSP, we decoded the route step by step. More specifically, for burma14 and ulysses22, we discarded nodes that have already been visited and selected the most likely node to visit based on our model output; for st70, we found that this greedy decoding failed to find good enough routes and instead applied beam search that expands the top 5 possible valid nodes at each step.

All models were implemented using PyTorch (Paszke et al., 2019) and PyTorch Geometric (Fey & Lenssen, 2019). Experiments were conducted on a machine with a single NVIDIA GeForce RTX 4090 GPU, a 32-core Intel Core i9-14900K CPU, and 64 GB of RAM running Ubuntu 24.04.

# D ADDITIONAL EXPERIMENTAL RESULTS

## D.1 DYCO-GNN WITH SAGECONV FOR DYNAMIC MAXCUT AND MIS

Table 5: Mean ApR on dynamic MaxCut. Values closer to 1 are better (↑). All methods use SAGEConv. The best result for each time budget is in bold. The first time each method surpasses the converged solution of static PI-GNN is highlighted.

|  | Method | \multicolumn{8}{c}{Time budget (epochs/seconds)} | | | | | | | |
| --- | --- | --- | --- | --- | --- | --- | --- | --- | --- |
|  |  | 50/0.06 | 100/0.12 | 200/0.23 | 300/0.34 | 500/0.56 | 1000/1.11 | 2000/2.21 | 3000/3.32 |
| Infectious | PI-GNN (static) | 0.97028 | 0.97353 | 0.97487 | 0.97540 | 0.97570 | 0.97591 | 0.97626 | 0.97636 |
|  | PI-GNN (warm) | 0.95685 | 0.96117 | 0.96356 | 0.96452 | 0.96396 | 0.96625 | 0.96841 | 0.96619 |
|  | DyCO-GNN (full) | **0.97771** | **0.97854** | **0.97880** | **0.97953** | **0.97952** | **0.97959** | **0.98051** | **0.98009** |
|  | Method | 50/0.06 | 100/0.12 | 200/0.24 | 300/0.36 | 500/0.59 | 1000/1.18 | 2000/2.36 | 3000/3.53 |
| UC Social | PI-GNN (static) | 0.98875 | **0.99601** | **0.99714** | **0.99727** | **0.99766** | **0.99768** | 0.99773 | 0.99782 |
|  | PI-GNN (warm) | 0.99292 | 0.99395 | 0.99440 | 0.99456 | 0.99451 | 0.99447 | 0.99437 | 0.99456 |
|  | DyCO-GNN (full) | **0.99515** | 0.99584 | 0.99595 | 0.99594 | 0.99593 | 0.99589 | **0.99856** | **0.99852** |
|  | Method | 50/0.14 | 100/0.29 | 200/0.57 | 300/0.85 | 500/1.41 | 1000/2.82 | 2000/5.63 | 3000/8.44 |
| DBLP | PI-GNN (static) | 0.89694 | 0.92001 | 0.93263 | 0.93688 | 0.94120 | 0.94487 | 0.94610 | 0.94659 |
|  | PI-GNN (warm) | 0.94865 | 0.95037 | 0.95249 | 0.95282 | 0.95336 | 0.95477 | 0.95337 | 0.95444 |
|  | DyCO-GNN (full) | **0.95877** | **0.95942** | **0.95981** | **0.95971** | **0.95965** | **0.95944** | **0.95934** | **0.95907** |

Table 6: Mean ApR on dynamic MIS. Values closer to 1 are better (↑). All methods use SAGEConv. The best result for each time budget is in bold. The first time each method surpasses the converged solution of static PI-GNN is highlighted.

|  | Method | \multicolumn{8}{c}{Time budget (epochs/seconds)} | | | | | | | |
| --- | --- | --- | --- | --- | --- | --- | --- | --- | --- |
|  |  | 50/0.06 | 100/0.12 | 200/0.23 | 300/0.34 | 500/0.56 | 1000/1.12 | 2000/2.22 | 3000/3.33 |
| Infectious | PI-GNN (static) | 0.79468 | 0.91679 | 0.95948 | 0.96362 | 0.96410 | 0.96476 | 0.96518 | 0.96518 |
|  | PI-GNN (warm) | **0.96725** | **0.96713** | **0.96727** | **0.96708** | **0.96701** | **0.96717** | 0.96715 | 0.96687 |
|  | DyCO-GNN (full) | 0.81767 | 0.96411 | 0.96540 | 0.96555 | 0.96592 | 0.96707 | **0.96746** | **0.96746** |
|  | Method | 50/0.06 | 100/0.12 | 200/0.24 | 300/0.36 | 500/0.60 | 1000/1.19 | 2000/2.37 | 3000/3.56 |
| UC Social | PI-GNN (static) | 0.71087 | 0.91961 | 0.96766 | 0.97126 | 0.97284 | 0.97398 | 0.97478 | 0.97516 |
|  | PI-GNN (warm) | **0.97607** | 0.97654 | 0.97665 | 0.97675 | 0.97677 | 0.97670 | 0.97673 | 0.97681 |
|  | DyCO-GNN (full) | 0.91199 | **0.97913** | **0.97838** | **0.97760** | **0.97761** | **0.97708** | **0.97698** | **0.97708** |
|  | Method | 50/0.14 | 100/0.28 | 200/0.57 | 300/0.85 | 500/1.40 | 1000/2.80 | 2000/5.60 | 3000/8.41 |
| DBLP | PI-GNN (static) | 0.60354 | 0.95227 | 0.98754 | 0.98950 | 0.99025 | 0.99081 | 0.99113 | 0.99130 |
|  | PI-GNN (warm) | 0.99357 | 0.99363 | 0.99364 | 0.99364 | 0.99367 | 0.99367 | 0.99367 | 0.99366 |
|  | DyCO-GNN (full) | **0.99551** | **0.99494** | **0.99465** | **0.99437** | **0.99426** | **0.99399** | **0.99386** | **0.99399** |

## D.2 RESULTS OF THE LAST CHECKPOINTS OF DYCO-GNN ON DYNAMIC TSP

Table 7: Mean ApR on dynamic TSP. Values closer to 1 are better ($\downarrow$). The last checkpoint was taken. "emb", "GNN", and "full" refer to applying SP to the embedding layer, GNN layers, and all layers, respectively. The best result for each time budget is in bold. The first time each method surpasses the converged solution of static PI-GNN is highlighted.

| | | Time budget (epochs/seconds) | | | | | | |
|---|---|---|---|---|---|---|---|---|
| | Method | 500/0.31 | 1000/0.62 | 2000/1.24 | 3000/1.86 | 5000/3.10 | 10000/6.20 | - |
| burma14 | PI-GNN (static) | 1.35521 | 1.14307 | 1.07890 | 1.06514 | 1.05002 | 1.04025 | - |
| | PI-GNN (warm) | 1.30116 | 1.29941 | 1.27008 | 1.28396 | 1.24094 | 1.16453 | - |
| | DyCO-GNN (emb) | 1.16825 | 1.15260 | 1.11613 | 1.08597 | 1.09302 | 1.08198 | - |
| | DyCO-GNN (GNN) | **1.13749** | **1.08812** | 1.06424 | 1.07396 | **1.05675** | **1.03002** | - |
| | DyCO-GNN (full) | 1.19695 | 1.15270 | **1.05980** | **1.04272** | 1.06671 | 1.08105 | - |
| | | Time budget (epochs/seconds) | | | | | | |
| | Method | 500/0.32 | 1000/0.64 | 2000/1.27 | 3000/1.90 | 5000/3.17 | 10000/6.36 | - |
| ulysses22 | PI-GNN (static) | 1.77516 | 1.54757 | 1.29137 | 1.21368 | **1.13068** | **1.10295** | - |
| | PI-GNN (warm) | **1.18274** | 1.17251 | 1.18645 | 1.18424 | 1.18650 | 1.22231 | - |
| | DyCO-GNN (emb) | 1.21109 | **1.16790** | **1.17498** | 1.17131 | 1.15733 | 1.13694 | - |
| | DyCO-GNN (GNN) | 1.32308 | 1.27132 | 1.18273 | **1.14293** | 1.15254 | 1.13845 | - |
| | DyCO-GNN (full) | 1.36858 | 1.33861 | 1.28760 | 1.22112 | 1.16052 | 1.12970 | - |
| | | Time budget (epochs/seconds) | | | | | | |
| | Method | 500/0.38 | 1000/0.76 | 2000/1.51 | 3000/2.27 | 5000/3.78 | 10000/7.57 | 20000/15.16 |
| st70 | PI-GNN (static) | 2.10292 | 2.11124 | 2.03993 | 1.94988 | 1.70642 | 1.55780 | 1.46919 |
| | PI-GNN (warm) | **1.48778** | 1.49261 | 1.47614 | 1.46951 | **1.44118** | 1.43268 | 1.43030 |
| | DyCO-GNN (emb) | 1.49505 | **1.49011** | **1.44635** | 1.45302 | 1.41491 | 1.38462 | 1.35803 |
| | DyCO-GNN (GNN) | 1.65917 | 1.58705 | **1.46435** | **1.43776** | 1.44108 | **1.35839** | **1.30869** |
| | DyCO-GNN (full) | 1.67305 | 1.68751 | 1.50680 | **1.46486** | **1.39795** | 1.37431 | 1.33007 |

## D.3 RESULTS OF NON-NEURAL BASELINES

Here we include the results of non-neural approaches built into NetworkX (Hagberg et al., 2008). Each entry of Tables 8, 9, and 10 is of the form ApR (wall clock time). For dynamic MIS, we consider the method in Boppana & Halldórsson (2006); the implementation in NetworkX only works for graphs with fewer than several hundred nodes without a recursion error, so we only report the result on Infectious.

Our method works the best except for dynamic MIS on Infectious and dynamic TSP on st70. We note that the method in Boppana & Halldórsson (2006) is specifically designed for solving MIS, and the implementation in NetworkX cannot handle graphs with more than several hundred nodes, whereas our method and the baselines have broader applicability and can easily handle graphs with more than 10,000 nodes. It has also been shown in Schuetz et al. (2022) that PI-GNN performs on par or better than Boppana & Halldórsson (2006) on random d-regular graphs. For TSP on st70, the result of the greedy approach is better than that of Gurobi, which could be an edge case. Nevertheless, our method improves upon the PI-GNN baselines by a large margin.

Table 8: Comparison with non-neural baseline on dynamic MaxCut. Values closer to 1 are better ($\uparrow$).

| Dataset | Cut-based greedy | PI-GNN (static) | PI-GNN (warm) | DyCO-GNN |
|---|---|---|---|---|
| Infectious | 0.96921 (5.6s) | 0.98888 (3.32s) | 0.97186 (0.56s) | 0.98947 (0.56s) |
| UC Social | 0.98083 (933.16s) | 0.99823 (3.53s) | 0.99519 (1.18s) | 0.99825 (0.24s) |
| DBLP | Fail to find a solution in 10 hours | 0.99095 (8.44s) | 0.99039 (8.44s) | 0.99319 (0.14s) |

Table 9: Comparison with non-neural baseline on dynamic MIS. Values closer to 1 are better ($\uparrow$).

| Dataset | Boppana & Halldórsson (2006) | PI-GNN (static) | PI-GNN (warm) | DyCO-GNN |
|---|---|---|---|---|
| Infectious | 0.93825(6.88s) | 0.87651 (3.33s) | 0.50845 (0.56s) | 0.88254 (1.12s) |

Table 10: Comparison with non-neural baseline on dynamic TSP. Values closer to 1 are better ($\downarrow$).

| Dataset | Cost-based greedy | PI-GNN (static) | PI-GNN (warm) | DyCO-GNN |
|---|---|---|---|---|
| burma14 | 1.19812 (<0.01s) | 1.03358 (6.21s) | 1.13860 (6.21s) | 1.01811 (6.21s) |
| ulysses22 | 1.24475 (<0.01s) | 1.08867 (6.37s) | 1.17228 (0.64s) | 1.08516 (6.37s) |
| st70 | 0.94963 (<0.01s) | 1.36945 (16.95s) | 1.34213 (16.95s) | 1.23655 (16.95s) |

## D.4 SENSITIVITY ANALYSIS ON SP PARAMETERS.

From the results reported in Tables 11 and 12, $0.2 \leq \lambda_{\text{shrink}} \leq 0.4$ and $\lambda_{\text{perturb}} = 0.1$ are good default choices. When $\lambda_{\text{shrink}}$ approaches 1, the ApRs will potentially be the highest early on, as this setting is close to a naive warm start; the ApRs may drop in this case when the number of epochs increases because the parameters closer to convergence on the previous snapshot without proper SP are not suitable for adaptation. We note that one should not confuse the performance with respect to epochs with the case in regular deep learning training: we are showing the number of epochs each snapshot is optimized for before adapting the parameters for the next snapshot, and the ApRs are computed across all snapshots. In general, a larger $\lambda_{\text{shrink}}$ gives better performance when the time budget is extremely tight, while a smaller $\lambda_{\text{shrink}}$ gives better performance when the time budget constraint is slightly relaxed. $\lambda_{\text{perturb}}$ has a smaller effect than $\lambda_{\text{shrink}}$. $\lambda_{\text{perturb}}$ is usually too aggressive; values like 0.1 and 0.01 typically give the best results.

Table 11: Mean ApR on dynamic MaxCut using UC Social achieved by DyCO-GNN with different SP parameters. Values closer to 1 are better ($\uparrow$).

| | | Time budget (epochs/seconds) | | | | | | | |
|---|---|---|---|---|---|---|---|---|---|
| $\lambda_{\text{shrink}}$ | $\lambda_{\text{perturb}}$ | 50/0.06 | 100/0.12 | 200/0.23 | 300/0.34 | 500/0.56 | 1000/1.12 | 2000/2.22 | 3000/3.33 |
| 0.2 | 0.1 | 0.99142 | 0.99747 | **0.99850** | **0.99862** | **0.99865** | **0.99872** | **0.99873** | **0.99875** |
| 0.4 | 0.1 | 0.99403 | 0.99724 | 0.99825 | 0.99841 | 0.99844 | 0.99848 | 0.99846 | 0.99848 |
| 0.6 | 0.1 | 0.99544 | **0.99805** | 0.99832 | 0.99830 | 0.99829 | 0.99807 | 0.99793 | 0.99782 |
| 0.8 | 0.1 | **0.99702** | 0.99738 | 0.99760 | 0.99744 | 0.99748 | 0.99742 | 0.99740 | 0.99740 |
| 0.4 | 1.0 | 0.98437 | 0.99459 | 0.99731 | 0.99777 | 0.99793 | 0.99793 | 0.99789 | 0.99785 |
| 0.4 | 0.1 | 0.99403 | 0.99724 | 0.99825 | **0.99841** | **0.99844** | **0.99848** | 0.99846 | **0.99848** |
| 0.4 | 0.01 | 0.99408 | **0.99732** | **0.99826** | 0.99836 | 0.99840 | 0.99847 | **0.99848** | 0.99846 |
| 0.4 | 0.001 | **0.99416** | **0.99732** | **0.99826** | 0.99835 | 0.99839 | 0.99845 | **0.99848** | 0.99845 |

Table 12: Mean ApR on dynamic MIS using UC Social achieved by DyCO-GNN with different SP parameters. Values closer to 1 are better ($\uparrow$).

| | | Time budget (epochs/seconds) | | | | | | | |
|---|---|---|---|---|---|---|---|---|---|
| $\lambda_{\text{shrink}}$ | $\lambda_{\text{perturb}}$ | 50/0.06 | 100/0.12 | 200/0.23 | 300/0.34 | 500/0.56 | 1000/1.12 | 2000/2.22 | 3000/3.33 |
| 0.2 | 0.1 | 0.48314 | 0.53766 | 0.68657 | 0.80957 | 0.88256 | 0.90916 | 0.91371 | 0.91607 |
| 0.4 | 0.1 | 0.54819 | 0.66505 | 0.80108 | 0.87436 | **0.91599** | **0.92047** | **0.92046** | **0.91987** |
| 0.6 | 0.1 | 0.70601 | **0.83711** | **0.90015** | **0.89537** | 0.88777 | 0.87972 | 0.87283 | 0.86969 |
| 0.8 | 0.1 | **0.84145** | 0.82497 | 0.80988 | 0.80543 | 0.80072 | 0.79774 | 0.79606 | 0.79350 |
| 0.4 | 1.0 | **0.59099** | **0.69377** | **0.81221** | 0.86738 | 0.90229 | 0.90993 | 0.90641 | 0.90533 |
| 0.4 | 0.1 | 0.54819 | 0.66505 | 0.80108 | **0.87436** | **0.91599** | **0.92047** | **0.92046** | 0.91987 |
| 0.4 | 0.01 | 0.54720 | 0.66718 | 0.80084 | 0.87241 | 0.91504 | 0.92001 | 0.91994 | 0.91921 |
| 0.4 | 0.001 | 0.54733 | 0.66713 | 0.80096 | 0.87235 | 0.91491 | 0.92020 | **0.92046** | **0.91994** |

918
919
920
921
922
923
924
925
926
927
928
929
930
931
932
933
934
935
936

## D.5 SNAPSHOT-LEVEL APRS BAR PLOTS

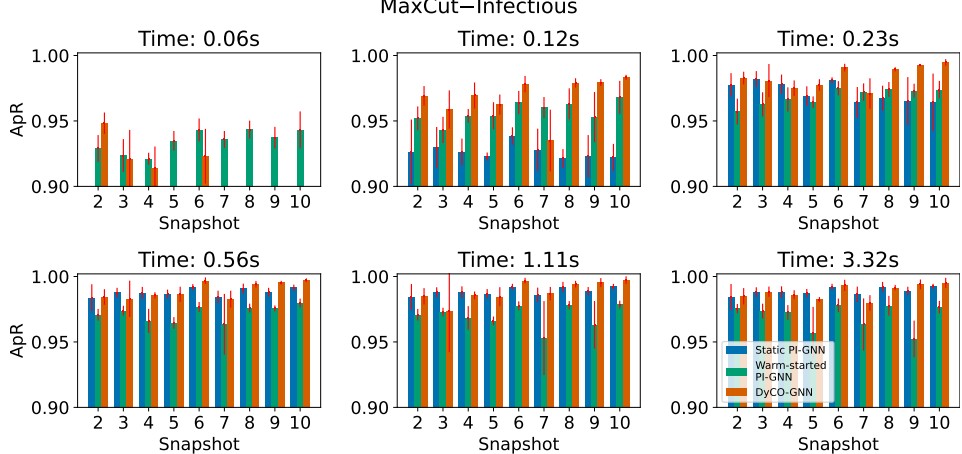

Figure 6: Snapshot-level ApRs on dynamic MaxCut instance (Infectious). All methods use GCNConv.

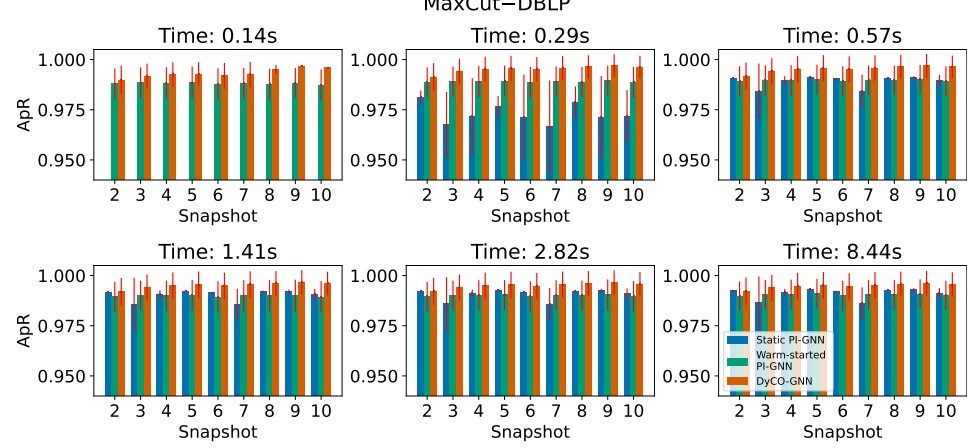

Figure 7: Snapshot-level ApRs on dynamic MaxCut instance (DBLP). All methods use GCNConv.

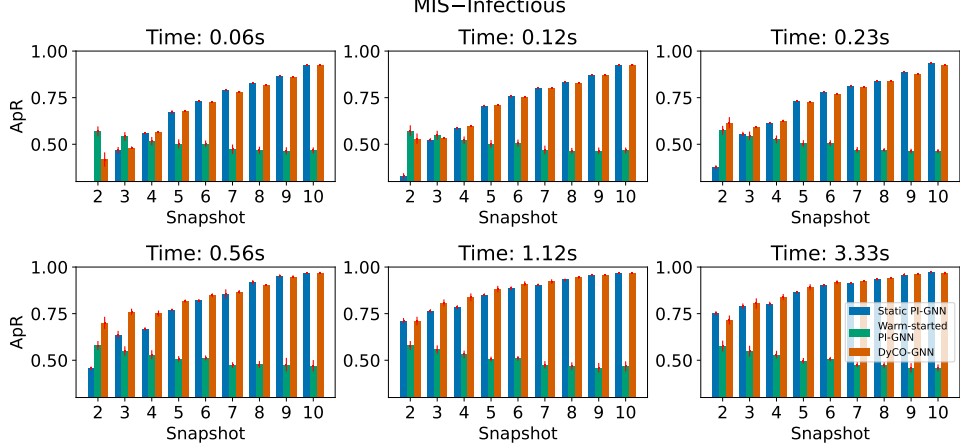

Figure 8: Snapshot-level ApRs on dynamic MIS instance (Infectious). All methods use GCNConv.

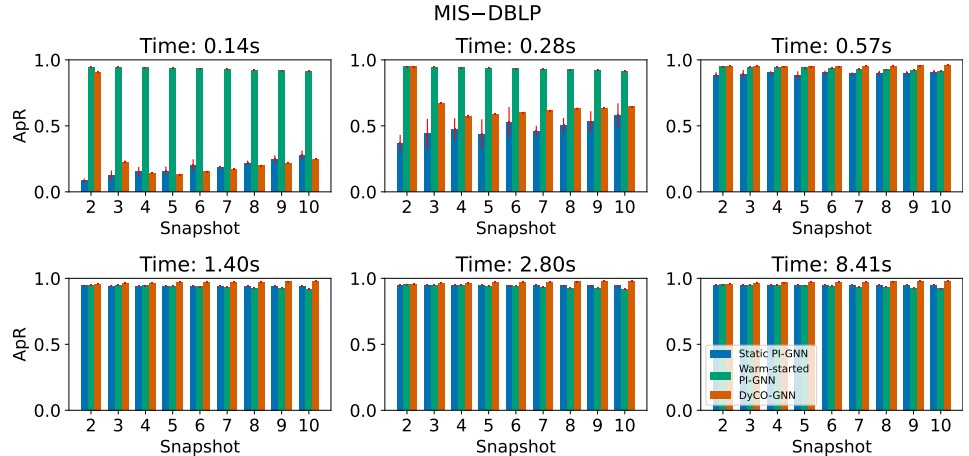

Figure 9: Snapshot-level ApRs on dynamic MIS instance (DBLP). All methods use GCNConv.

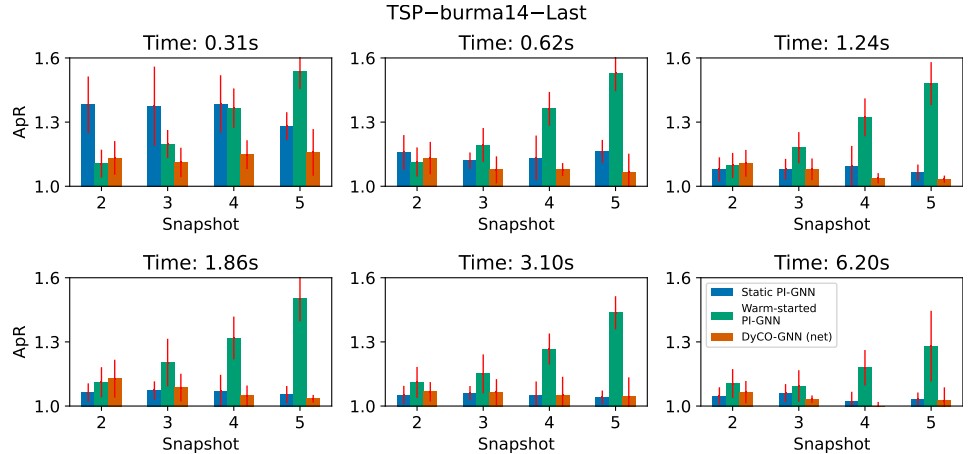

Figure 10: Snapshot-level ApRs on dynamic TSP instance (burma14). The last checkpoint was taken.

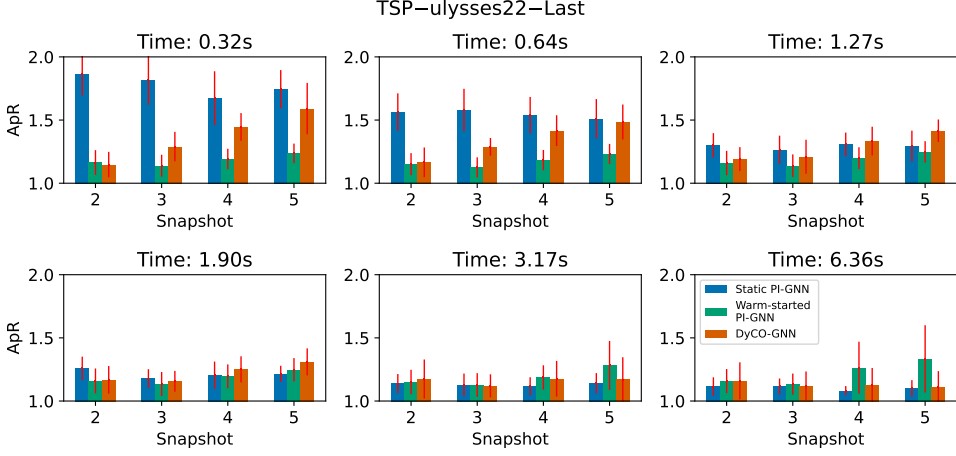

Figure 11: Snapshot-level ApRs on dynamic TSP instance (ulysses22). The last checkpoint was taken.

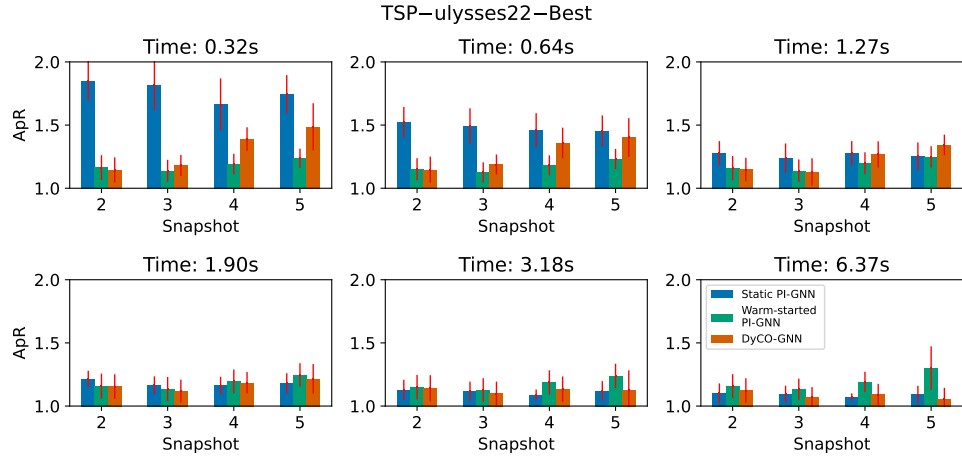

Figure 12: Snapshot-level ApRs on dynamic TSP instance (ulysses22). The best checkpoint was taken.

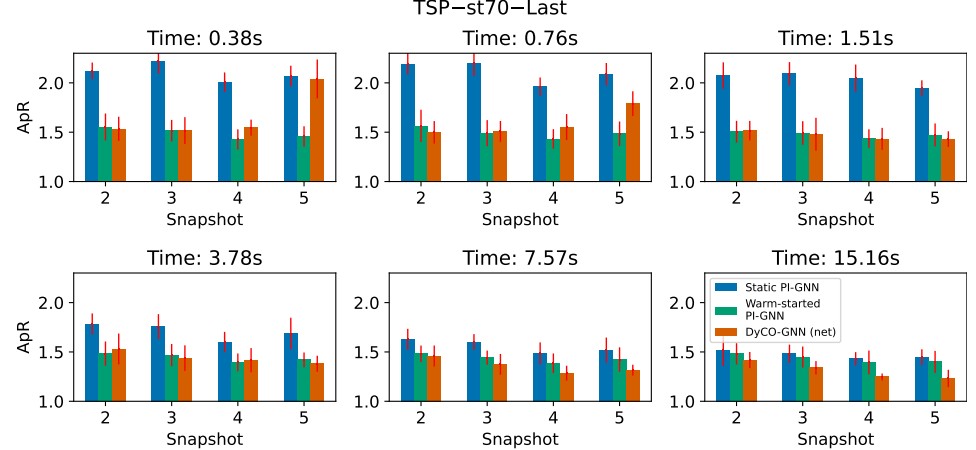

Figure 13: Snapshot-level ApRs on dynamic TSP instance (st70). The last checkpoint was taken.

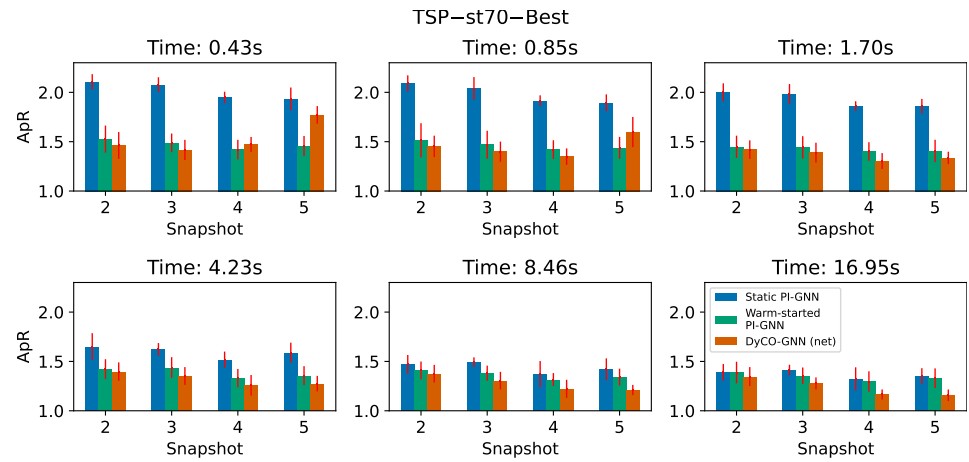

Figure 14: Snapshot-level ApRs on dynamic TSP instance (st70). The best checkpoint was taken.

# E    PROOF AND EXTENSION OF THEOREM 1

## E.1    THE GOEMANS-WILLIAMSON (GW) ALGORITHM

Let $G = (V, E)$ be an unweighted graph with $|V| = n$ nodes, and let $L \in \mathbb{R}^{n \times n}$ be its Laplacian matrix. Let $c^*$ denote the maximum achievable cut size on this graph. Our goal is to attain this maximum cut size.

Consider the following QUBO reformulation of the MaxCut problem on this graph.

$$\min \sum_{(i,j) \in E} \frac{1 - x_i x_j}{2}, \qquad \text{s.t. } x_i \in \{-1, 1\}. \tag{2}$$

Goemans & Williamson (1995) showed that the best approximation ratio (of at least 0.87856 times the optimal value) for the MaxCut problem can be attained by proceeding as follows. We first reformulate the above QUBO problem as the Semidefinite Program (SDP):

$$\max_X \ \frac{1}{4}\text{Tr}(LX), \qquad \text{s.t. } X \succeq 0, \ \text{diag}(X) = 1 \,, \tag{3}$$

where $\text{Tr}(\cdot)$ denotes the trace of a matrix. Let $X_{\text{SDP}}^*$ denote the optimal solution obtained from solving the SDP. This solution is within the feasible set $\mathcal{X} = \{X \mid X \succeq 0, \ \text{diag}(X) = 1\}$; this is the set of all positive definite matrices with ones on the diagonal. Once the SDP is solved, the GW algorithm proceeds by performing a "randomized rounding" on an eigen-decomposition of the solution $X_{\text{SDP}}^*$ to produce a binary vector representing a cut.

In more detail, consider the eigen-decomposition $X_{\text{SDP}}^* = A\Lambda A^T$ where $A \in \mathbb{R}^{n \times n}$ is an orthogonal matrix and $\Lambda$ is a diagonal matrix, with the eigenvalues of $X_{\text{SDP}}^*$ on its diagonal. Define the matrix $Y = A\Lambda^{\frac{1}{2}}$; this can be viewed as an "embedding" of the solution produced by the SDP. In it, each row $Y_i$ corresponds to node $i$, which needs to be mapped to $+1$ or $-1$ (this will be the label of the node, which in turn will determines the two sets produced by the cut). To attain these node labels, in the rounding cut, the GW rounding algorithm samples a random vector $r \sim \mathcal{N}(0, I_n)$ (where $I_n$ denotes the $n \times n$ identity matrix), and defines the cut to be $x_i = \text{sign}(Y_i^T r)$.

## E.2    PROOF OF THEOREM 1

*Proof.* First, note that the noise $Z$ has full support on the space $\mathbb{S}^n$ consisting of all $n \times n$ symmetric matrices, by definition. Let its distribution be denoted by $\nu$.

Next, we show that the projection $\text{Proj}_{\mathcal{X}}$ is continuous. This is because the space $\mathbb{S}^n$ is a finite-dimensional real Hilbert space under the Frobenius inner product $< A, B >= \text{Tr}(AB)$. The feasible set $\mathcal{X} \subset \mathbb{S}^n$ is the intersection of the cone of positive semidefinite matrices and an affine subspace defined by linear constraints $\text{diag}(X) = 1$. Both the positive semidefinite cone and the affine subspace are closed and convex, and therefore $\mathcal{X}$ is a closed, convex subset of a Hilbert space. By the Hilbert Projection Theorem (Bauschke & Combettes, 2017, Theorem 3.12) the Euclidean projection onto any closed convex subset is continuous. Therefore, the projection function $\text{Proj}_{\mathcal{X}}$ is continuous.

Now define the function $f : \mathbb{S}^n \to \mathcal{X}$ as $f(Z) := \text{Proj}_{\mathcal{X}}(X_0 + \lambda Z)$. This function determines the distribution of $\mathcal{X}_\lambda$ through the distribution of $Z$. Formally, let $\mu_\lambda = f_\# \nu$ be the pushforward measure on $\mathcal{X}$, i.e., the distribution of $X_\lambda$.

We now apply the following standard fact about pushforward measures: Let $\nu$ be a probability measure on a space $Y$, let $f : Y \to X$ be a measurable function, and let $\mu = f_\# \nu$ be the pushforward measure on $X$. Then for any measurable subset $A \subseteq X$, we have $\mu(A) = \nu(f^{-1}(A))$. In particular, if $\nu(f^{-1}(A)) > 0$, then $\mu(A) > 0$.

Back to our problem setting, note that by assumption, $\mathcal{C}_{\text{opt}} \subset \mathcal{X}$ has positive Lebesgue measure. As the measure $\nu$ of $Z$ has full support and is absolutely continuous with respect to the Lebesgue measure on $\mathbb{S}^n$, and since the projection function $\text{Proj}_{\mathcal{X}}$ is continuous, the pre-image $\text{Proj}_{\mathcal{X}}^{-1}(\mathcal{C}_{\text{opt}}) \subset \mathbb{S}^n$ has positive $\nu$-measure for some $\lambda > 0$. Therefore,

$$\mathbb{P}(R(X_\lambda) = c^*) = \mu_\lambda(C_{\text{opt}}) > 0 \,.$$

Finally, note that by the assumption $X_0 \notin \mathcal{C}_{opt}$, we have $\mathbb{P}(R(X_0) = c^*) = 0$. This completes the proof. $\qquad\square$

### E.3 Extension

Theorem 1 shows that the probability of finding the optimal cut strictly improves if we introduce noise pre-rounding; i.e., on a given, fixed solution of the SDP. The next corollary shows that the same result holds if we introduce noise in the initial solution of the SDP.

We use the same notation as in Theorem 1. That is, we let the feasible set of the SDP be denoted by $\mathcal{X} = \{X \mid X \succeq 0, \ \mathrm{diag}(X) = 1\}$, and $\mathrm{Proj}_{\mathcal{X}}(\cdot)$ is projection onto this feasible set $\mathcal{X}$. Denote the GW rounding step by $R : \{\mathcal{X}, \Omega\} \to \{0, 1, \ldots, c^*\}$, where $\Omega$ is the random seed set of the cut plane and $c^*$ is the maximum achievable cut size. We again let $\mathcal{C}_{opt} = \{X \in \mathcal{X} : \ \mathbb{P}_{\Omega}(R(X, \omega) = c^*) > 0\}$; this is the set of SDP solutions that have positive probability of yielding the optimal cut after the GW randomized rounding step.

**Corollary 1.** *Let $X_0 \in \mathcal{X}$ be the initial solution of the GW SDP. Define $X_\lambda := Proj_{\mathcal{X}}[X_0 + \lambda Z]$, where $\lambda \in \mathbb{R}_{\geq 0}$, and $Z$ is a symmetric random matrix sampled from the Gaussian Orthogonal Ensemble. Let $\Pi_{SDP}(X)$ denote the SDP solver starting from a feasible solution $X$, and assume that it is locally continuous. Assume that the set $C_{opt}$ has positive Lebesgue measure in $\mathcal{X}$, that $\Pi(X_0) \notin \mathcal{C}_{opt}$. Then, there exists a $\lambda > 0$ such that*

$$\mathbb{P}_{\Omega, Z}(R(\Pi(X_\lambda), \omega) = c^*) > \mathbb{P}_{\Omega}(R(\Pi(X_0), \omega) = c^*) = 0 \ .$$

The proof is straightforward: given the assumption that the SDP solver $\Pi(\cdot)$ is locally continuous, that means that small perturbations of the initial SDP solution $X_0$ map to perturbations of the solution $\Pi(X_0)$. Consequently, the results of Theorem 1 are applicable. We note that SDP solvers for solving MaxCut, such as interior-point methods, are locally continuous under mild assumptions; e.g. Shapiro (1988).

## F Use of Large Language Models

Large language models were used for editing purposes only.

