# OpenReview forum: "Learning for Dynamic Combinatorial Optimization without Training Data"
_ICLR.cc/2026/Conference — ICLR 2026 Conference Withdrawn Submission_

### Official Review · Reviewer_Hptt · 2025-10-31

**Soundness:** 3
**Presentation:** 3
**Contribution:** 2
**Rating:** 4
**Confidence:** 4

**Summary:**

The paper introduces DyCO-GNN a technique for approximating the solution to dynamic CO problems using a GNN.  The DyCO-GNN is regularized at each epoch and then perturbed with noise.  This improves the online performance of the method as demonstrated by evaluations against one competing method PI-GNN.  The paper considers various ways of splitting dynamic graph data into epochs across tasks such as Max Cut, MIS, and TSP. They indicate superior performance to variations of PI-GNN.

**Strengths:**

The paper has many strengths.  I personally like tackling the dynamic graph optimization domain as most of the work in this area is about the static setting.  The choice to not train on an independent dataset is a well done problem framing choice.  It would have simply complicated the investigation.  The gains over PI-GNN static appear to be consistent across datasets and tasks.  The spread of datasets and CO tasks is extensive and incredibly thorough.

**Weaknesses:**

I have some questions about theorem 1, see questions.  As with every paper in this domain, I must ask how the neural approach compares to simple baselines like gurobi.  Or even the truly simplest random walk, simulated annealing, etc.  Without a solver baseline of any form it becomes exceptionally difficult to judge the performance of the models.

**Questions:**

I find the theorem 1 to be written in a rather confusing manner.  The probability of finding the optimal cut is greater than 0 is what the theorem says?  Does not strike me as being particularly interesting.  The randomized rounding is randomized anyway.  You can always add mean zero noise and the expected cut value will not change.  So sure maybe there’s some probability it could improve, is that all the theorem is communicating?  If so, I’d suggest removing it as it doesn’t really add any value to the paper.

---

### Official Review · Reviewer_F5eo · 2025-11-02

**Soundness:** 2
**Presentation:** 2
**Contribution:** 2
**Rating:** 2
**Confidence:** 4

**Summary:**

The paper proposes a methodology for solving QUBO combinatorial problems in the dynamic setting where one wants a solution to the problem over a sequence of graphs that displays temporal dependencies. The paper proposes using a specific shrinkage and perturbation technique to the parameters of the model instead of warm starting between snapshots and shows how this achieves superior results over static baselines and warm start techniques.

**Strengths:**

- The direction of solving qubo problems over dynamic graphs is an interesting and underexplored one.
- The paper provides results for the proposed method in multiple qubo problems.

**Weaknesses:**

- I have some issues with the experimental setion of the paper. I provide a brief list:
     - I think a simple fast non-neural baseline should be included in the main tables. I noticed in the appendix that some of your non-neural algorithms timed out. However, for example the Boppana algorithm is an approximation algorithm and not greedy heuristic. Its performance is guaranteed, but it can be much slower than a heuristic. You may want to consider something like the heuristic for MIS that was used in 1. If those time out, it may also be because of NetworkX and it might just be easier to reimplement them in PyG. If your neural network can run on these graphs then surely a simple MIS/maxcut algorithm should be fine too.
     - There are stronger (static) neural baselines than the ones this paper is considering. I think what needs to be clearer here is how the proposed apporach also compares to state of the art neural (static) baselines on those problems (see for example [2] for MIS/TSP, or [3,4] for Maxcut. For non-neural maxcut you may want to consider a fast lowrank sdp approach such as the one in [5].
- It is rather unclear to me how 4.3 lends theoretical support to your idea.
     - First, let me preface this by saying that I haven't looked at the proof too closely but I'm not sure how you establish $ P(R(X_\lambda)=c^*) = \mu_\lambda(\mathcal{C}_\text{opt}) $.  How is the probability that the rounded noisy embeddings give exactly optimal cuts equal to the pushforward measure of $\mathcal{C} _{\text{opt}}.$  Isn't the latter just the pushforward of matrices who can round to $c$ with some probability? Also why do you need the Gaussian Orthogonal Ensemble? It's also not clear why you necessarily get positive measure after noise + projection but maybe I'm missing something. Finally, there are some notational issues (e.g. $C _\text{opt}$ instead of caligraphic).
     - Proof aside, even if the statement is true, I don't see how it can say anything about the setting you are using SP. First of all, there is no shrinkage as far as I can tell in that theorem even though you make heavy use of it. Furtheremore, I don't see how the SDP solutions to the goemans williamson algorithm can in any way be related to the way you apply SP in the neural network setting. Maybe there is a vague analogy that can be drawn between nonlinearities and rounding but this seems too far from anything you're actually doing.
     - Even if the result did transfer somehow to the neural network setting, it ultimately makes a claim about a nonzero probability event. We have no bound on this probability, so for all we know it could be exponentially small. It's hard to see what the theorem actually buys us. So t's hard to judge whether this claim is meaningful or nontrivial.

Overall, I think this is an interesting direction and the paper explores some interesting ideas but I think the execution currently is not good enough to merit acceptance. I will reconsider after the rebuttal.



1. Toenshoff, Jan, et al. "Graph neural networks for maximum constraint satisfaction." Frontiers in artificial intelligence 3 (2021): 580607.
2. Li, Yang, et al. "Fast t2t: Optimization consistency speeds up diffusion-based training-to-testing solving for combinatorial optimization." Advances in Neural Information Processing Systems 37 (2024): 30179-30206.
3. Tönshoff, Jan, et al. "One model, any CSP: graph neural networks as fast global search heuristics for constraint satisfaction." arXiv preprint arXiv:2208.10227 (2022).
4. Yau, Morris, et al. "Are graph neural networks optimal approximation algorithms?." Advances in Neural Information Processing Systems 37 (2024): 73124-73181.
5. Wang, Po-Wei, Wei-Cheng Chang, and J. Zico Kolter. "The mixing method: low-rank coordinate descent for semidefinite programming with diagonal constraints." arXiv preprint arXiv:1706.00476 (2017).

**Questions:**

see above

---

### Official Review · Reviewer_F5vC · 2025-11-04

**Soundness:** 2
**Presentation:** 2
**Contribution:** 2
**Rating:** 2
**Confidence:** 3

**Summary:**

This paper introduces DyCO-GNN, an unsupervised learning framework for solving Dynamic Combinatorial Optimization (DCO) problems. The proposed method builds on the instance-specific optimization paradigm of PI-GNN, which requires no offline training data. The authors first demonstrate that a naive warm-start baseline (using the solution from snapshot $t-1$ to initialize snapshot $t$) converges quickly but to suboptimal solutions, likely due to overconfidence and local minima. The key contribution is to replace this naive warm-start with a "Shrink and Perturb" (SP) initialization strategy. The authors evaluate DyCO-GNN on dynamic versions of MaxCut, MIS, and TSP, showing that it consistently outperforms both static (from-scratch) and warm-started PI-GNN baselines, achieving high-quality solutions up to 3-60x faster.

**Strengths:**

1. The paper is clearly written and well-structured.


2. The paper tackles a practical and important problem (DCO) that has been relatively neglected by the learning-based CO community.


3. The empirical investigation is thorough. The method is validated on three distinct NP-hard problems (MaxCut, MIS, TSP) using multiple real-world dynamic graph datasets.

**Weaknesses:**

1. There may be an overclaim in line 53: "To the best of our knowledge, our work is the first to apply machine learning to DCO problems". To my knowledge, there are also some works studying how to solve dynamic/stochastic COPs, such as [1,2].


2. It is unclear why we do not use learned models to solve each snapshot of dynamic COPs. The importance of using UL to solve dynamic COPs is not well illustrated.

3. The experiment involves three dynamic COPs, but for dynamic TSP, the problem size is too small. It is unclear whether the UL method for each testing instance can only be applied to simple dynamic COPs.

[1] Neural Combinatorial Optimization for Stochastic Flexible Job Shop Scheduling Problems, AAAI 2025.

[2] Dynamic Capacitated Vehicle Routing Problem with Stochastic Requests Using Deep Reinforcement Learning, SMC 2024.

**Questions:**

1. Why not use learned models in [1,2,3,4] to solve dynamic TSP or MIS? I believe the learned model can effectively solve each snapshot of the dynamic COPs, resulting in high-quality solutions.

2. Please also clarify the weaknesses.

[1] POMO: Policy Optimization with Multiple Optima for Reinforcement Learning, Neurips2020.

[2] Neural Combinatorial Optimization with Heavy Decoder: Toward Large Scale Generalization, Neurips2023.

[3] BQ-NCO: Bisimulation Quotienting for Efficient Neural Combinatorial Optimization, Neurips2023.

[4] Fast T2T: Optimization Consistency Speeds Up Diffusion-Based Training-to-Testing Solving for Combinatorial Optimization, Neurips2024.

---

### Note · Authors · 2025-11-22

I have read and agree with the venue's withdrawal policy on behalf of myself and my co-authors.